# Typical Sulfonamide Antibiotics Removal by Biochar-Amended River Coarse Sand during Groundwater Recharge

**DOI:** 10.3390/ijerph192416957

**Published:** 2022-12-16

**Authors:** Rui Liu, Hechun Yu, Xiaoshu Hou, Xiang Liu, Erping Bi, Wenjing Wang, Miao Li

**Affiliations:** 1China Urban Construction Design & Research Institute Co., Ltd., Beijing 100120, China; 2Nanchang Institute of Environmental Science Co., Ltd., Nanchang 330000, China; 3Chinese Academy of Environmental Planning, Beijing 100012, China; 4School of Environment, Tsinghua University, Beijing 100084, China; 5Key Laboratory of Water Resources and Environment Engineering, School of Water Resources and Environment, China University of Geosciences (Beijing), Beijing 100083, China

**Keywords:** sulfonamide antibiotics, biochar, river coarse sand, groundwater recharge

## Abstract

The high porosity of medium-coarse sand (MCS) layers in groundwater recharge areas presents a high environmental risk. Sulfamethoxazole (SMX) and trimethoprim (TMP) are two common sulfonamide antibiotics in surface water that have a high propensity to migrate into groundwater. In this study, four biochars were prepared and biochar-amended soil aquifer treatment (SAT) columns were constructed to remove SMX and TMP. Batch experiments demonstrated that the sorption isotherms conformed to the Freundlich model. The maximum adsorptions of biochars prepared at 700 °C were 54.73 and 67.62 mg/g for SMX and 59.3 and 73.38 mg/g for TMP. Electrostatic interaction may be one of the primary mechanisms of adsorption. The column experiments showed that the SMX and TMP removal rate of the biochar-amended SAT was as high as 96%, while that of the MCS SAT was less than 5%. The addition of biochar greatly improved the retention capacity of the pollutants in the MCS layer in the groundwater recharge area and effectively reduced environmental risk.

## 1. Introduction

With increased urbanization, providing sufficient water supply to meet demand has become increasingly challenging. Moreover, with greater impervious areas, the amount of extracted groundwater greatly exceeds the amount of recharge, which poses a serious threat to the water supply. Therefore, a number of surface-water recharge projects have been carried out in a high-porosity infiltration basin for groundwater replenishment [1]. However, recharge with surface water carries certain environmental risks. Sulfamethoxazole (SMX) and trimethoprim (TMP) are typical sulfonamide antibiotics that have frequently been detected at high concentrations in surface-water environments [2,3]. SMX and TMP have strong migration potential. Particularly in high-porosity soils, these pollutants readily migrate into aquifers, resulting in potential risks to groundwater quality [4,5]. 

Researchers have recently studied the removal of sulfonamide antibiotics during recharge. These results demonstrate that soil aquifer treatment (SAT) using adsorptive media (e.g., biochar, activated carbon, carbon nanotubes, cross-linked polymer resins), advanced oxidation, and membrane technologies can effectively remove such trace toxic substances in water and soil [6,7,8,9]. However, advanced oxidation and membrane technologies are limited in practical application due to initial investment, high operating cost, and stringent technical requirements [10].

Biochar is an accessible and environmentally friendly adsorbent. Due to its structural characteristics, such as high porosity and large specific surface area, biochar demonstrates a strong adsorption efficiency of antibiotics [11]. Furthermore, biochar can be used as a stabilizer to control the migration and transformation of soil contaminants and reduce environmental risks. The adsorption of antibiotics by biochar differs based on source, calcination temperature, and environmental factors such as pH, coexisting ions, and soil characteristics [12,13,14]. 

Biochar is also recorded as a potential adsorbent to remove fatty acids, heavy metals, color, and other toxins from surface and groundwater [15,16]. A few researchers reported that different sources of biochar or modified biochar will give different rates of adsorption [17].

In this study, biochar-amended SATs were constructed to remove SMX and TMP, selected as representative sulfonamide antibiotics. In the SAT system, medium-coarse sand (MCS) collected from a high infiltration area of a typical surface-water recharge site was used as the fill medium. Biochars prepared from corn and wheat straw at carbonization temperatures of 500 and 700 °C were used as additives to improve the efficiency of SMX and TMP removal. By simulating the actual environment, this study demonstrates an effective approach toward preventing SMX, TMP, and other similar pollutants from entering groundwater during surface-water recharge. 

## 2. Materials and Methods

### 2.1. Preparation of Biochar

The biochars were prepared from corn straw and wheat straw. The preparation procedure of biochar was prepared according to the strategy noted in reference [18]. The thermogravimetric analyses of corn straw and wheat straw are shown in Appendix A. Firstly, the biomass was washed and dried, then smashed by a blender into smaller particles and sieved through a 40-mesh sieve. Then, the pretreated biomass was calcined at 500 °C and 700 °C for 6 h with a heating rate of 5 °C/min in a muffle furnace. The mass ratio before and after pyrolysis was the yield of the biochar. The prepared biochars were labeled as C500, C700, W500, and W700, respectively. They were washed with deionized water and dried at 100 °C. At last, the biochars were ground to powder, passed through a 100-mesh sieve and stored in desiccator.

### 2.2. Material Characterization

The apparent surface area and pore volume of the adsorbents were obtained using the nitrogen adsorption–desorption isotherms conducted on a specific surface area analyzer (ASAP2460, Micromeritics Instrument Corp, Norcross, GA, USA) at 77 K. The zeta potentials of the materials were measured on a Zetasizer Nano (Delsa Nano C, Beckman Coulter Inc.). C, H, and N were determined using a Heraeus CHN-O-RAPID (Hanau, Germany) elemental analyzer.

### 2.3. Batch Experiments

Batch experiments were used to investigate the adsorption interaction of SMX and TMP to biochars. The experimental background solution was 0.01 mol/L CaCl_2_ to maintain ionic strength. A total of 10 mg biochar was placed into a 50 mL tube. CaCl_2_ and SMX or TMP were added from the stock solution. They were placed in a shaker with a speed of 150 rpm/min at 25 °C in the dark conditions. Two parallel samples were set. Then, tubes were centrifuged at 9000 rmp/min for 10 min, and the solution was filtered with the filter membrane (0.22 μm). The concentrations of SMX and TMP were determined by high-performance liquid chromatography (Appendix A).

The adsorbed amount of SMX/TMP, *Qe* (mg/g), was determined according to Equation (1):(1)$$Q_{e}=\frac{\left(C_{0}-C_{e}\right)V}{M}$$
where *C*_0_ and *C_e_* are the initial and equilibrium concentrations of SMX/TMP (mg/L) in filtrate, respectively, *V* is the volume of solution (L), and *M* is the dry mass of biochar used (g).

### 2.4. Biochar-Amended SAT

#### 2.4.1. Packed Soil Column Set-Up

Each column consisted of a polymethyl methacrylate cylinder (L = 30 cm, i.d. = 5 cm). The bottom–up mediums were the support layer (3 cm), filling medium (20 cm), water distribution layer (1 cm), and overflow layer (6 cm). The schematic diagram of the packed soil column is shown in Figure 1. The filling medium was the mixture of biochar and MSC collected from the riverbed in the high infiltration area and the proportion of biochar was 2.08%. Before being filled into the cylinder, the sand and biochar were mixed in a vortex mixer for 30 min. 

#### 2.4.2. Packed Soil Column Experiments

Prior to pumping any polluted water, the packed columns were inlet with deionized water until the deionized water rose to the top of the columns and the volume of water was recorded. Then, the columns were washed with deionized water to remove soluble impurities and easily migrated with dissolved organic matter and particulate matter from the column. Next, a tracer test was conducted in each column with NaBr (40 mg/L) pumped to determine the porosity. Then, after the cleaning period to remove the residual bromine ions, the simulated water was pumped into the columns. The recharge experiment was divided into two stages. In the first stage, 20 μg/L (1000 times the actual determined result) SMX and TMP were added for 45 d. In the second stage, 200 μg/L (10,000 times the measured concentration) SMX and TMP were added. The water was pumped at a flow rate of 0.6 mL/min.

## 3. Results and Discussion

### 3.1. Characterization of Biochars

As shown in Table 1, for the biochars prepared from wheat straw, the content of C increased with the increase in pyrolysis temperature, while for biochars prepared from corn straw the content of C decreased, which was mainly due to the relatively large proportion of ash. With the increase in the pyrolysis temperature, amorphous carbon gradually changed into a dense aromatic ring structure with a large amount of C [19]. The contents of H and O gradually decreased with the increase in the pyrolysis temperature, which indicated that the oxygen-containing functional groups in the biochar decreased. The variation in N content of the biochars was related to the instability of the nitrogen-containing functional groups [20]. With the increase in pyrolysis temperature, the aromaticity and hydrophilicity of the biochar decreased, represented by H/C and O/C, respectively. This may have been due to the decrease in oxygen-containing functional groups. The polarity index of biochar was represented by (O + N)/C, and decreased with the increase in pyrolysis temperature, while the yield decreased and ash content increased [21].

As shown in Table 2, due to the transformation of aliphatic carbon into aromatic rings of graphene structure, the pore volume and specific surface area increased with increasing carbonization temperature and the micropores increased. Except C500, the microporous area of biochar was significantly larger than the outer surface area, accounting for more than 75% of the total specific surface area [22].

The zeta potential of biochars was analyzed, as shown in Figure 2. In the range of pH 2−11, the zeta potential of the four biochars decreased rapidly and then increased to different degrees. At pH of the experiment environment (about 6), the surface of the biochar had a negative charge. The zero-point potential measured for the four biochars was at about pH = 2. This is consistent with the results of a previous study [23].

### 3.2. The Adsorption of SMX/TMP on Biochars

#### 3.2.1. Kinetics of SMX and TMP Adsorption on Biochars

Figure 3 shows the adsorption amounts of SMX/TMP on biochars as a function of contact time. At first, the amount of adsorbed SMX/TMP increased rapidly. However, due to the limited adsorption sites on the biochar, the adsorption rate became slower with adsorption time. The amount of adsorbed SMX/TMP reached the adsorption equilibrium at approximately 72 h on the biochar prepared at 500 °C and at 168 h on the biochar prepared at 700 °C. Compared with the biochar prepared at 500 °C, this may have been due to richer micropores on the biochar prepared at 700 °C and more time being needed for the interaction between the micropores and SMX/TMP. 

To understand the process better, the dual-chamber first-order kinetics model was employed to fit the adsorption data. The equation is as follows:(2)$$Q_{e}=\left[f_{1}\left(1-e^{-k_{1}t}\right)+f_{2}(1-e^{-k_{2}t}\right]$$
where *k*_1_ is the adsorption rate constant of fast adsorption, h^−1^; *k*_2_ is the adsorption rate constant of slow adsorption, h^−1^; and *f*_1_ + *f*_2_ = 1.

The kinetics constants are presented in Table 3. The dual-chamber first-order kinetics model fitted the experiment data better with a higher correlation coefficient (*R*^2^ = 0.94–0.99). The fast adsorption rate constant *k*_1_ ranges from 2.46/h to 6.23/h, while the slow adsorption rate constant *k*_2_ ranges from 0.03/h to 0.13/h. This is at the same level as the adsorption rate of phenanthrene in soil [24], and *k*_1_ is 18.9–207 times larger than *k*_2_. At the beginning of the adsorption, fast adsorption dominated, and then the contribution of slow sorption gradually increased until the adsorption equilibrium was reached. Among the four biochars, the proportion of slow adsorption in the adsorption process of SMX for W500 was the smallest and about 18%, while the proportions of W500 and C500 were about 27%. In addition, the adsorption of TMP on biochars prepared at 500 °C and 700 °C showed similar results to the adsorption of SMX at 700 °C, and the fast adsorption process dominated.

#### 3.2.2. Adsorption Isotherm by Biochars

There are several adsorption isotherm models such as the linear, Freundlich, Langmuir, Brouers–Sotolongo (B–S), Temkin model, and Dubinin–Ashtakhov (DA) isotherms. The Langmuir model and the Freundlich model are two common models frequently used to describe the relationship [25]. The models are as follows:Freundlich: *Q_e_* = *K_f_*·(*C_e_*)*^n^*(3)
Langmuir: *Q_e_* = (*Q_m_·K_L_·C_e_*)/(1 + *K_L_·C_e_*)(4)
where *n* is a empirical constant and a measure of non-linearity adsorption and *K_f_
*((mmol/kg)(L/mmol)*^n^*) is a constant indicating the relative adsorption capacity of the adsorbent [26]; *K_L_* (L/kg) is a constant characterizing the adsorption surface strength, which is related to the adsorption bonding energy; and *Q_m_* is the maximum adsorption amount (mg/g) [27].

Figure 4 shows the adsorption of SMX/TMP on the biochars, and the constants of the Langmuir and Freundlich model fitting curves are listed in Table 4. The Freundlich isotherm equation (*R*^2^ = 0.88–0.98) can fit the adsorption of SMX and TMP on biochar better than the Langmuir isotherm equation (*R*^2^ = 0.73–098) with a higher correlation coefficient, indicating that the adsorption took place on a heterogeneous surface. With the initial concentrations of SMX and TMP increased, the adsorption amount increased, while the growth rate of adsorption amounts slowed down until it reached stability, mainly because the adsorption sites on the surface of biochar were limited. With the increase in the pyrolysis temperature, the adsorption quantities increased and the maximum adsorption amounts (*Q_m_*) of the biochars prepared at 700 °C were 2.31–4.72 times that of the biochars prepared at 500 °C. This may have been due to the microporous nature inside the biochar, and that the specific surface area increased at higher pyrolysis temperatures [28]. In addition, the value of *n* decreased as the pyrolysis temperature increased, and the nonlinearity of adsorption isotherms gradually increased, indicating that more inhomogeneous “vitreous”, “hard carbon”, or highly concentrated adsorption potentials were formed on the surface of biochar, and the adsorption energy was higher for these adsorption sites [29,30].

Compared with previous studies [31,32], the adsorption amounts of SMX and TMP by the biochar in this study were greater, which may be related to the longer pyrolysis time and the removal of ash by hydrochloric acid soaking during the preparation of biochar [33]. The adsorption quantity of TMP by the same biochar was more than SMX. At the experimental pH 6.5, SMX^−^ and TMP^+^ were the dominant species, while the surfaces of the biochars had a negative charge. Moreover, there was electrostatic attraction between the biochar and TMP, while electrostatic repulsion between the biochar and SMX worked. Hence, electrostatic interaction was one of the dominant mechanisms for TMP adsorption on the adsorbents [34].

### 3.3. The Removal of SMX/TMP for Biochar-Amended SAT

#### 3.3.1. Parameters Fitting for Biochar-Amended SAT

The parameters of each column are listed in Table 5. In order to simulate the concentration of SMX/TMP in the effluent of the experimental column at different time points, a suitable soil solute transport simulation fit needs to be selected, and the transport fit parameters are mainly the hydrodynamic dispersion coefficient and the apparent permeation rate of the experimental column. The equations are listed in the Appendix A.

#### 3.3.2. Removal of SMX and TMP for Soil Columns 

The penetration experimental data of the bromine ion were fitted with CXTFIT 2.0 software (USDA-ARS). The flow rate was 0.0828 cm/min and the dispersion coefficient was 0.0251 cm^2^/min with a high correlation coefficient (*R*^2^ = 0.999). The retention factors of SMX in columns 2 and 4 were obtained by fitting the variation curves of SMX using CXTFIT software. The removal rates of TMP by column 2 and 4 and SMX/TMP by column 3 and column 5 were also calculated.

In the first stage, the removals of SMX and TMP were strong in all soil columns, and the removal rates reached 100%. With the recharge quantity increasing in the second stage, the penetration curves of SMX in column 2 and column 4 were obtained, as shown in Figure 5. SMX was not detected in the first 240 h. SMX was detected until columns 2 and 4 were running for 290 h and 347 h, respectively. The measured data and simulation curves fitted by CXTFIT software are shown in Figure 5. The *R*^2^ values of the two curves were 0.983 and 0.994, respectively, indicating a high degree of fitting between measured and simulated values. The retention factors for columns 2 and 4 were 115.2 and 135.9, respectively. The retention factor is an important parameter describing the adsorption–desorption of contaminants in the column. Compared with adsorption capacity, the retention factor was related to maximum adsorption capacity and the fast chamber adsorption constant. The biochar with greater adsorption capacity and faster adsorption rate had a stronger hindering effect on SMX. The penetration process was slower for column 4 with a high adsorption capacity and the C/C_0_ values were less than 1 for all experimental columns. This was probably due to the inhomogeneity and surface roughness of the filled media particles with biochar addition, and SMX was trapped in the low flow rate zone of the pore media. The hydraulic retention time was about 4 h. The decay constants of SMX on biochar reported in the literature were all greater than 4 h [35,36], and the microbial degradation of SMX was weak.

The removal rates of SMX and TMP by column 3 and column 5 are shown in Figure 6. In the second stage, the removal rate of SMX in the column 1 was less than 5%, while in columns 3 and 5, the removal rates were more than 97%. Furthermore, the removal rates of TMP in columns 2 and 4 were more than 82%, while the removal rates in columns 3 and 5 were more than 96%. This was related to the adsorption capacity. The maximum adsorption capacities of TMP on W700 and C700 were 59.3 mg/g and 73.38 mg/g, respectively, and 2.31 and 3.65 times the maximum adsorption capacities on W500 and C500. Furthermore, a longer time was needed to detect it in the effluent, indicating that SAT with biochar prepared at a higher temperature can more effectively remove pollutants from the recharge water source and reduce the risk. Additionally, the maximum adsorption capacities of TMP on C700 and W700 were significantly higher than those of SMX, respectively. However, columns 3 and 5 showed similar removal efficiencies for SMX and TMP. This illustrated that besides adsorption capacity, there were other factors influencing the penetration process. In the actual environment, the concentration of SMX was about a few dozen nanograms per liter. The contaminant could be effectively removed by the larger adsorption capacity of biochar-amended SAT, and the potential risk of sulfonamide antibiotics and their similar contaminants in the recharge process could be reduced.

#### 3.3.3. The Adsorption Capacity of SMX in the Experimental Columns

The calculations of the adsorption amount of SMX are in the Appendix A. The adsorption capacities of SMX were 1103.19 μg/g and 1325.83 μg/g for column 2 and column 4, respectively. The adsorption capacity was correlated with the maximum adsorption capacity in the batch experiments, and the difference between the two adsorption capacities was one order of magnitude. It is possible that the short residence time in the recharge process resulted in insufficient reaction time between SMX and the biochar to reach the maximum adsorption capacity. The concentrations of SMX and TMP in the actual environment are tens of nanograms per liter, and the adsorption capacity of biochar in the biochar-amended SAT was several hundred or even thousand micrograms per gram. This indicates that the biochar-amended SAT has a great adsorption potential, and the pollutants could be removed in the recharge process.

## 4. Conclusions

In this study, four biochars were prepared and the adsorption effects on SMX and TMP were investigated. Due to high specific surface area and total pore volume, the adsorption capacity of biochar for SMX and TMP increased with pyrolysis temperature. Due to the influence of pH, the adsorption of TMP was greater than SMX. In addition, biochar-amended SATs were constructed, improving the retention of SMX and TMP. Retention factors varied with the adsorption capacity and fast chamber adsorption rate constants of the biochars. The removal rates of SMX and TMP in the biochar-amended SATs ranged up to 96%. Based on the high adsorption potential in the media simulating the actual environment, these pollutants can be removed during groundwater recharge. 

## Figures and Tables

**Figure 1 ijerph-19-16957-f001:**
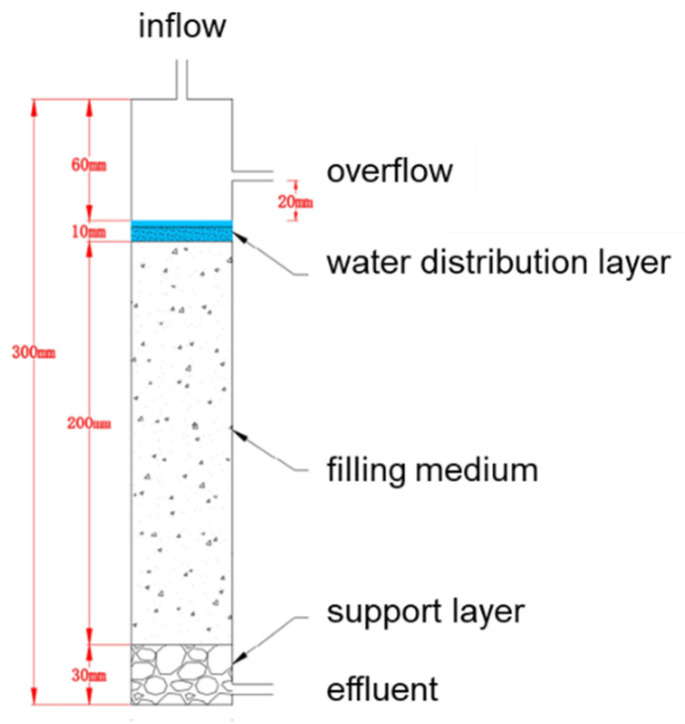
Schematic diagram of the packed soil column.

**Figure 2 ijerph-19-16957-f002:**
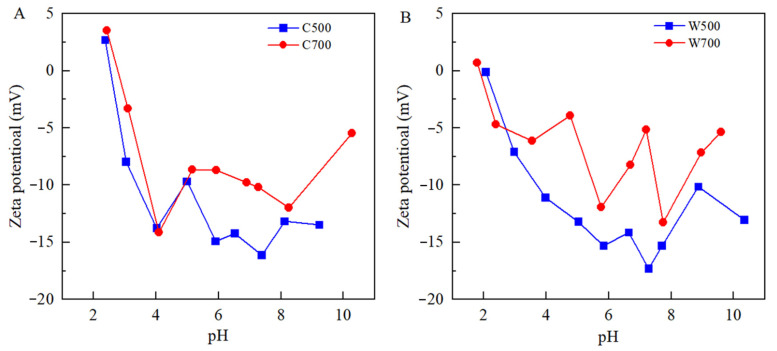
Zeta potential of corn biochars (**A**) and wheat biochars (**B**).

**Figure 3 ijerph-19-16957-f003:**
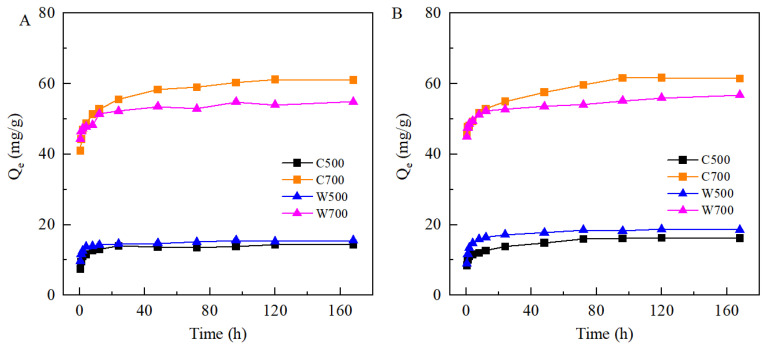
Effect of contact time on the adsorption of SMX (**A**) and TMP (**B**) (C_0_ (SMX/ TMP) = 30 mg/L for the biochar prepared at 500 °C and 60 mg/L for the biochar prepared at 700 °C).

**Figure 4 ijerph-19-16957-f004:**
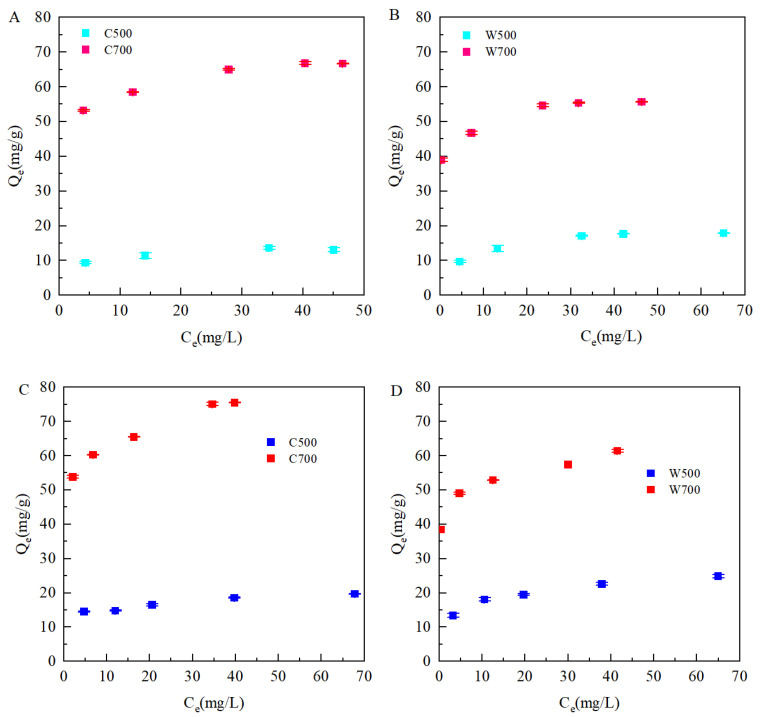
Adsorption for SMX (**A**,**B**)/TMP (**C**,**D**) on biochars.

**Figure 5 ijerph-19-16957-f005:**
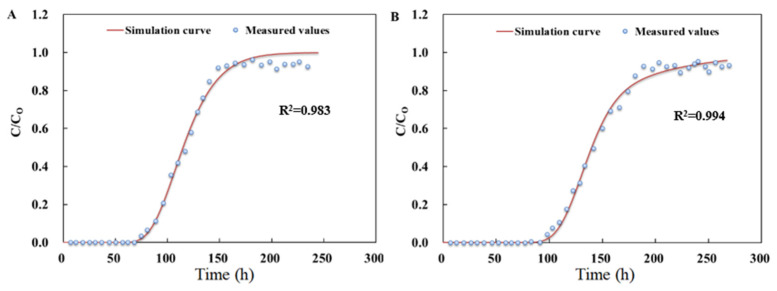
The penetration experimental data of SMX ((**A**) column 2; (**B**) column 4).

**Figure 6 ijerph-19-16957-f006:**
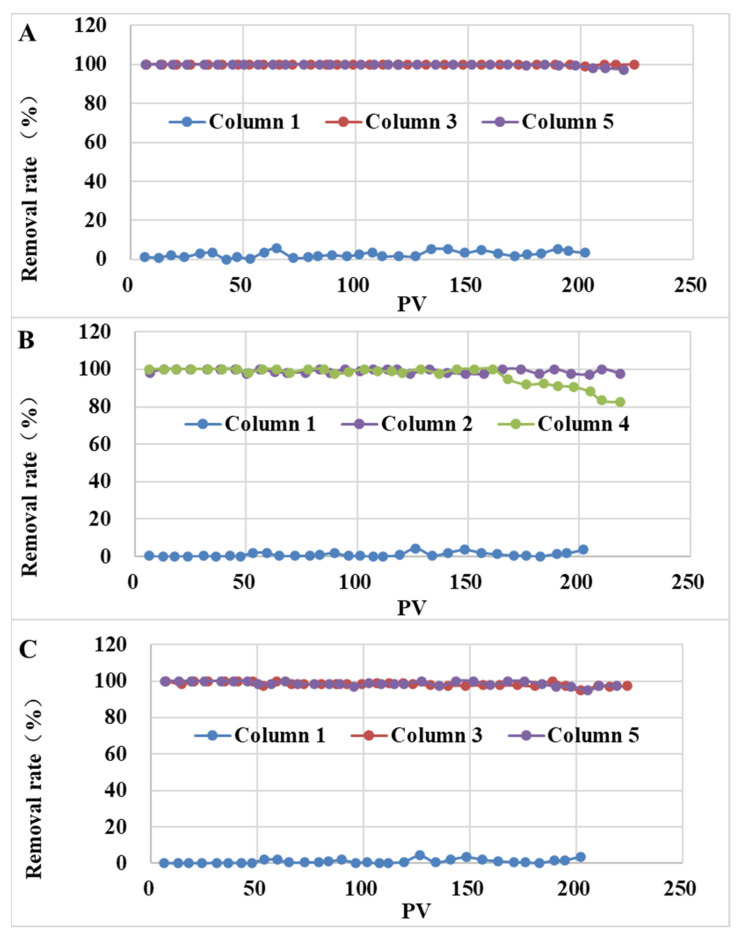
The removal rates of each experimental column ((**A**): SMX; (**B**) and (**C**): TMP).

**Table 1 ijerph-19-16957-t001:** Elemental composition, ratio, and yield of biochars.

Biochars	C/%	H/%	O/%	N/%	Ash Content/%	O/C	H/C	(O+N)/C	Yield/%
W500	63.50	2.16	17.75	0.89	15.70	0.28	0.03	0.29	23.79
W700	57.28	1.04	11.07	0.59	30.03	0.19	0.02	0.20	16.48
C500	67.44	2.68	24.13	1.62	4.13	0.36	0.04	0.38	22.64
C700	72.67	1.92	17.73	1.03	6.65	0.24	0.03	0.26	15.01

**Table 2 ijerph-19-16957-t002:** BET analysis of biochars.

Biochar	SSA ^a^ (m^2^/g)	S_mic_ ^b^ (m^2^/g)	ESSA ^c^ (m^2^/g)	V_t_ ^d^ (cm^3^/g)	V_mic_ ^e^ (cm^3^/g)	PS ^f^ (nm)
W500	461.00	393.90	67.09	0.28	0.18	2.43
W700	532.80	410.40	122.40	0.37	0.19	2.77
C500	369.00	175.58	193.46	0.30	0.09	3.24
C700	598.60	453.74	144.90	0.42	0.22	2.80

Note: ^a^: specific surface area; ^b^: micropore area; ^c^: external specific surface area; ^d^: pore volume; ^e^: micropore volume; ^f^: pore size.

**Table 3 ijerph-19-16957-t003:** Model results of adsorption kinetics data.

		Dual-Chamber First-Order Kinetics Model					
		*Q_e_* (mg/g)	*f* _1_	*f* _2_	*k*_1_ (h^−1^)	*k*_2_ (h^−1^)	*R* ^2^
SMX	W500	15.21	0.82	0.18	2.83	0.09	0.96
	C500	13.97	0.72	0.28	2.46	0.13	0.98
	W700	54.17	0.85	0.15	6.28	0.06	0.96
	C700	60.39	0.75	0.25	4.34	0.05	0.98
TMP	W500	18.29	0.67	0.33	2.60	0.10	0.99
	C500	16.38	0.67	0.33	2.51	0.03	0.99
	W700	55.54	0.87	0.13	5.17	0.04	0.94
	C700	61.68	0.78	0.23	6.23	0.03	0.99

**Table 4 ijerph-19-16957-t004:** Adsorption models of SMX/TMP on biochars.

		Langmuir Model			Freundlich Model		
		*K_L_*	*Q_m_*	*R* ^2^	*K_f_*	*n*	*R* ^2^
SMX	W500	0.23	19.27	0.98	8.87	0.18	0.87
	W700	7.87	54.73	0.73	42.41	0.07	0.92
	C500	0.43	14.33	0.97	8.09	0.13	0.93
	C700	0.71	67.62	0.80	46.28	0.10	0.98
TMP	W500	0.34	25.63	0.90	10.75	0.21	0.96
	W700	1.01	59.3	0.83	41.58	0.1	0.98
	C500	0.55	20.1	0.92	11.29	0.14	0.88
	C700	1.01	73.38	0.82	48.18	0.12	0.98

**Table 5 ijerph-19-16957-t005:** Parameters of each column.

No.	Filling Medium	Pore Volume (mL)	Porosity	Effective Pore Volume (mL)	Specific Tetention (%)	Column Volume (mL)
1	MCS ^a^	157	0.40	23.1	42.5	392.5
2	MCS + C500	140	0.357	21.6	35.7	392.5
3	MCS + C700	141	0.359	20.8	35.9	392.5
4	MCS + W500	142	0.362	19.4	36.2	392.5
5	MCS + W700	143	0.364	22.1	36.4	392.5

Note: ^a^: medium-coarse sand.

## Data Availability

The associated dataset for the study is available upon request to the corresponding author.

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
