# Peer review of "Typical Sulfonamide Antibiotics Removal by Biochar-Amended River Coarse Sand during Groundwater Recharge"

_ijerph, 2022, doi:10.3390/ijerph192416957_

Round 1

Reviewer 1 Report

Introduction 

1) After line 54, perhaps can be added this information. 

" Biochar is also recorded as a potential adsorbent to remove fatty acids, heavy metals, color, and other toxins from surface and groundwater (A &B). A few researchers reported that different sources of biochar or modification of biochar will be given different rates of adsorption (C)

A-Selective adsorption and recovery of volatile fatty acids from fermented landfill leachate by activated carbon process.

B-Adsorption efficiency and isotherms of COD and color using limestone and zeolite adsorbents

C-Ammoniacal nitrogen and COD removal from stabilized landfill leachate using granular activated carbon and green mussel (Perna Viridis) shell powder as a composite adsorbent

2.0 Materials and methods.

2.4.1 Packed soil columns set-up (Schematic diagram is required)

3.0 Result

i)Table 1.- Simple explanation the calculation of % yeild.

ii)Figure 1.- How about zeta potential (charges) of a sulfonamide antibiotic? If you not run in your experiment, perhaps you can cite previous researchers. 

iii) Equation (2) typo error. Should be "Freundlich".

iv) Figure 4- How fit the measured values and simulation? R2 is required. 

Reviewer 2 Report

The work presented consists of the evaluation of biochars in the retention of sulfonamide type contaminants for cleaning contaminated water. Although the work is interesting, it is difficult to find the novelty when there are multiple studies that replicate the methodologies used in this work, simply with other raw materials to develop biochars. If the novelty of the work was to obtain biochars from new raw materials, it would be expected that the raw material would be at least preliminarily characterized, of which there is nothing in this work. There is also no evaluation of the pyrolysis conditions, in order to optimize the biochar yield.

On the other hand, the analytical methods used are mentioned, in which neither the methodology nor the model or configuration of the instrument is described. 

-          In biochar preparation: In my opinion, there is a lack of a deeper analysis of the raw material (before pyrolysis). Essentially to know its elemental composition, proximate and ultimate analysis. It seems to me that a thermogravimetric analysis indicating the stability of the carbonaceous skeleton is also missing.

Line 65-66: “Firstly, the biomass was washed and dried at 100 ℃” Why to that temperature? There is any reference or ASTM norm?  Did you measure the pH of the resultant biochar?

Line 67-68: Why were chosen those temperatures? And those reaction times? If you don’t have a TGA characterization of the raw material, how do you choose these variables? How do you maximize the solid yield? (char)

Line 68: Did you use any gas inert flow? Which one?

Line 69-70: Why the biochars were washed? Did you measure the pH in the washing water? Again, why use that temperature in drying?  There is not any error associated to this experimental measuring?

In general, is not clear the obtaining of biochars. It seems like random variables were used for char obtaining without a previous study about the efficiency of the calcination process with this raw material. It would be expected that an optimization study would have been performed to obtain a specific pore size for the application to be studied, however, this is missing in the research, leaving us with the impression that a biocarbon is obtained under arbitrary conditions, which is used without evaluating how to maximize the solid yield, nor optimizing the pore size to maximize adsorption.

Line 86: “The concentrations of SMX and TMP were determined by high performance liquid chromatography” Where is described the method? Which instrument? Column? Wavelength? Also are necessary the calibration curves with the analytical parameters in Supplementary Information.

Line 117 “with the increasing of pyrolysis temperature”. Why do you talk about pyrolysis if none of inert gas has been described flowing  in methodology?

3.1 Characterization of biochars: only results description, no discussion about the behavior according to temperature and elemental analysis.

It would have been interesting to perform a SEM analysis to see the surface morphology of the biochar obtained at different temperatures.

Why type of pyrolysis regime did you use? Fast, slow? Why?

Why do you not perform a FTIR analysis before and after of pollutant adsorption?

What about the TOC in the treated water?

What about the Recovery and Regeneration of Adsorbent??
